# Submodular + Concave

**Siddharth Mitra**
School of Engineering & Applied Science
Yale University
New Haven, CT 06520–8292
siddharth.mitra@yale.edu

**Moran Feldman**
Department of Computer Science
University of Haifa
Haifa 3498838 , Israel
moranfe@cs.haifa.ac.il

**Amin Karbasi**
School of Engineering & Applied Science
Yale University
New Haven, CT 06520–8292
amin.karbasi@yale.edu

## Abstract

It has been well established that first order optimization methods can converge to the maximal objective value of concave functions and provide constant factor approximation guarantees for (non-convex/non-concave) continuous submodular functions. In this work, we initiate the study of the maximization of functions of the form $F(x) = G(x) + C(x)$ over a solvable convex body $P$, where $G$ is a smooth DR-submodular function and $C$ is a smooth concave function. This class of functions is a strict extension of both concave and continuous DR-submodular functions for which no theoretical guarantee is known. We provide a suite of Frank-Wolfe style algorithms, which, depending on the nature of the objective function (i.e., if $G$ and $C$ are monotone or not, and non-negative or not) and on the nature of the set $P$ (i.e., whether it is downward closed or not), provide $1 - 1/e$, $1/e$, or $1/2$ approximation guarantees. We then use our algorithms to get a framework to smoothly interpolate between choosing a diverse set of elements from a given ground set (corresponding to the mode of a determinantal point process) and choosing a clustered set of elements (corresponding to the maxima of a suitable concave function). Additionally, we apply our algorithms to various functions in the above class (DR-submodular + concave) in both constrained and unconstrained settings, and show that our algorithms consistently outperform natural baselines.

## 1 Introduction

Despite their simplicity, first-order optimization methods (such as gradient descent and its variants, Frank-Wolfe, momentum based methods, and others) have shown great success in many machine learning applications. A large body of research in the operations research and machine learning communities has fully demystified the convergence rate of such methods in minimizing well behaved convex objectives [Bubeck, 2015, Nesterov, 2018]. More recently, a new surge of rigorous results has also shown that gradient descent methods can find the global minima of specific non-convex objective functions arisen from non-negative matrix factorization [Arora et al., 2012], robust PCA [Netrapalli et al., 2014], phase retrieval [Chen et al., 2019b], matrix completion [Sun and Luo, 2016], and the training of wide neural networks [Du et al., 2019, Jacot et al., 2018, Allen-Zhu et al., 2019], to name a few. It is also very well known that finding the global minimum of a general non-convex function is indeed computationally intractable [Murty and Kabadi, 1987]. To avoid such impossibility results, simpler goals have been pursued by the community: either developing algorithms that can escape

saddle points and reach local minima [Ge et al., 2015] or describing structural properties under which reaching a local minimizer ensures optimality [Sun et al., 2016, Bian et al., 2017b, Hazan et al., 2016]. In the same spirit, this paper quantifies a large class of non-convex functions for which first-order optimization methods provably achieve near optimal solutions.

More specifically, we consider objective functions that are formed by the sum of a continuous DR-submodular function $G(x)$ and a concave function $C(x)$. Recent research in non-convex optimization has shown that first-order optimization methods provide constant factor approximation guarantees for maximizing continuous submodular functions Bian et al. [2017b], Hassani et al. [2017], Bian et al. [2017a], Niazadeh et al. [2018], Hassani et al. [2020a], Mokhtari et al. [2018a]. Similarly, such methods find the global maximizer of concave functions. However, the class of $F(x) = G(x) + C(x)$ functions is strictly larger than both concave and continuous DR-submodular functions. More specifically, $F(x)$ is not concave nor continuous DR-submodular in general (Figure 1 illustrates an example). In this paper, we show that first-order methods provably provide strong theoretical guarantees for maximizing such functions.

The combinations of continuous submodular and concave functions naturally arise in many ML applications such as maximizing a regularized submodular function [Kazemi et al., 2020a] or finding the mode of distributions [Kazemi et al., 2020a, Robinson et al., 2019]. For instance, it is common to add a regularizer to a D-optimal design objective function to increase the stability of the final solution against perturbations [He, 2010, Derezinski et al., 2020, Lattimore and Szepesvari, 2020]. Similarly, many instances of log-submodular distributions, such as determinantal point processes (DPPs), have been studied in depth in order to sample a diverse set of items from a ground set [Kulesza, 2012, Rebeschini and Karbasi, 2015, Anari et al., 2016]. In order to control the level of diversity, one natural recipe is to consider the combination of a log-concave (e.g., normal distribution, exponential distribution and Laplace distribution) [Dwivedi et al., 2019, Robinson et al., 2019] and log-submodular distributions [Djolonga and Krause, 2014, Bresler et al., 2019], i.e., $\Pr(\mathbf{x}) \propto \exp(\lambda C(\mathbf{x}) + (1 - \lambda)G(\mathbf{x}))$ for some $\lambda \in [0, 1]$. In this way, we can obtain a class of distributions that contains log-concave and log-submodular distributions as special cases. However, this class of distributions is strictly more expressive than both log-concave and log-submodular models, e.g., in contrast to log-concave distributions, they are not uni-modal in general. Finding the mode of such distributions amounts to maximizing a combination of a continuous DR-submodular function and a concave function. The contributions of this paper are as follows.

- Assuming first-order oracle access for the function $F$, we develop the algorithms: GREEDY FRANK-WOLFE (Algorithm 1) and MEASURED GREEDY FRANK-WOLFE (Algorithm 2) which achieve constant factors approximation guarantees between $1 - 1/e$ and $1/e$ depending on the setting, i.e. depending on the monotonicity and non-negativity of $G$ and $C$, and depending on the constraint set having the down-closeness property or not.

- Furthermore, if we have access to the individual gradients of $G$ and $C$, then we are able to make the guarantee with respect to $C$ *exact* using the algorithms: GRADIENT COMBINING FRANK-WOLFE (Algorithm 3) and NON-OBLIVIOUS FRANK-WOLFE (Algorithm 4). These results are summarized and made more precise in Table 1 and Section 3.

- We then present experiments designed to use our algorithms to smoothly interpolate between contrasting objectives such as picking a diverse set of elements and picking a clustered set of elements. This smooth interpolation provides a way to control the amount of diversity in the final solution. We also demonstrate the use of our algorithms to maximize a large class of (non-convex/non-concave) quadratic programming problems.

**Related Work.** The study of discrete submodular maximization has flourished in the last decade through far reaching applications in machine learning and and artificial intelligence including viral marketing [Kempe et al., 2003], dictionary learning [Krause and Cevher, 2010], sparse regression [Elenberg et al., 2016], neural network interoperability [Elenberg et al., 2017], crowd teaching [Singla et al., 2014], sequential decision making [Alieva et al., 2021], active learning [Wei et al., 2015], and data summarization [Mirzasoleiman et al., 2013]. We refer the interested reader to a recent survey by Tohidi et al. [2020] and the references therein.

Recently, Bian et al. [2017b] proposed an extension of discrete submodular functions to the continuous domains that can be of use in machine learning applications. Notably, this class of (non-convex/non-concave) functions, so called continuous DR-submodular, contains the continuous multilinear ex-

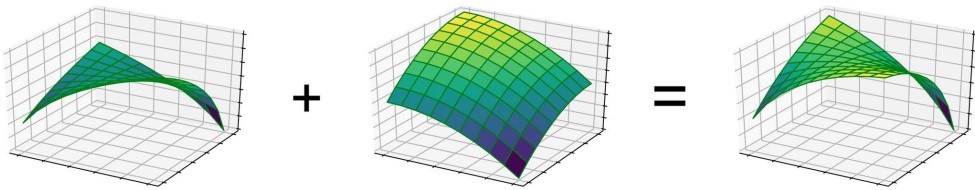

Figure 1: LEFT: (continuous) DR-submodular softmax extension. MIDDLE: concave quadratic function. RIGHT: sum of both.

tension of discrete submodular functions Călinescu et al. [2011] as a special case. Continuous DR-submodular functions can reliably model revenue maximization [Bian et al., 2017b], robust budget allocation [Staib and Jegelka, 2017], experimental design [Chen et al., 2018], MAP inference for DPPs [Gillenwater et al., 2012, Hassani et al., 2020b], energy allocation [Wilder, 2018b], classes of zero-sum games [Wilder, 2018c], online welfare maximization and online task assignment [Sadeghi et al., 2020], as well as many other settings of interest.

The research on maximizing continuous DR-submodular functions in the last few years has established strong theoretical results in different optimization settings including unconstrained [Niazadeh et al., 2018, Bian et al., 2019], stochastic Mokhtari et al. [2018a], Hassani et al. [2017], online [Chen et al., 2018, Zhang et al., 2019, Sadeghi and Fazel, 2019, Raut et al., 2021], and parallel models of computation [Chen et al., 2019a, Mokhtari et al., 2018b, Xie et al., 2019, Ene and Nguyen, 2019].

A different line of works study the maximization of discrete functions that can be represented as the sum of a non-negative monotone submodular function and a linear function. The ability to do so is useful in practice since the linear function can be viewed as a soft constraint, and it also has theoretical applications as is argued by the first work in this line [Sviridenko et al., 2017] (for example, the problem of maximization of a monotone submodular function with a bounded curvature can be reduced to the maximization of the sum of a monotone submodular function and a linear function). In terms of the approximation guarantee, the algorithms suggested by Sviridenko et al. [2017] were optimal. However, more recent works improve over the time complexities of these algorithms [Feldman, 2021, Harshaw et al., 2019, Ene et al., 2020], generalize them to weakly-submodular functions [Harshaw et al., 2019], and adapt them to other computational models such as the data stream and MapReduce models [Kazemi et al., 2020b, Ene et al., 2020].

## 2 Setting and Notation

Let us now formally define the setting we consider. Fix a subset $\mathcal{X}$ of $R^n$ of the form $\prod_{i=1}^{n} \mathcal{X}_i$, where $\mathcal{X}_i$ is a closed range in $\mathbb{R}$. Intuitively, a function $G\colon \mathcal{X} \to \mathbb{R}$ is called (continuous) *DR-submodular* if it exhibits diminishing returns in the sense that given a vector $\mathbf{x} \in \mathcal{X}$, the increase in $G(\mathbf{x})$ obtained by increasing $x_i$ (for any $i \in [n]$) by $\varepsilon > 0$ is negatively correlated with the original values of the coordinates of $\mathbf{x}$. This intuition is captured by the following definition. In this definition $\mathbf{e}_i$ denotes the standard basis vector in the $i^{\text{th}}$ direction.

**Definition 2.1.** *A function $G\colon \mathcal{X} \to \mathbb{R}$ is DR-submodular if for every two vectors $\mathbf{a}, \mathbf{b} \in \mathcal{X}$, positive value $k$ and coordinate $i \in [n]$ we have $G(\mathbf{a} + k\mathbf{e}_i) - G(\mathbf{a}) \geq G(\mathbf{b} + k\mathbf{e}_i) - G(\mathbf{b})$ whenever $\mathbf{a} \leq \mathbf{b}$ and $\mathbf{a} + k\mathbf{e}_i, \mathbf{b} + k\mathbf{e}_i \in \mathcal{X}$.*[1]

It is well known that when $G$ is continuously differentiable, then it is DR-submodular if and only if $\nabla G(\mathbf{a}) \geq \nabla G(\mathbf{b})$ for every two vectors $\mathbf{a}, \mathbf{b} \in \mathcal{X}$ that obey $\mathbf{a} \leq \mathbf{b}$. Furthermore, for the sake of simplicity we assume in this work that $\mathcal{X} = [0,1]^n$.

We are interested in the problem of finding the point in some convex body $P \subseteq [0,1]^n$ that maximizes a given function $F\colon [0,1]^n \to \mathbb{R}$ that can be expressed as the sum of a DR-submodular function $G\colon [0,1]^n \to \mathbb{R}$ and a concave function $C\colon [0,1]^n \to \mathbb{R}$. To get meaningful results for this problem, we need to make some assumptions. Here we describe three basic assumptions that we make throughout the paper. The quality of the results that we obtain improves if additional assumptions are made, as is described in Section 3.

---

[1]Throughout the paper, inequalities between vectors should be understood as holding coordinate-wise.

| Algorithm | $G$ | $C$ | $P$ | $\alpha$ | $\beta$ |
|---|---|---|---|---|---|
| Greedy Frank-Wolfe | Monotone & non-neg. | Monotone & non-neg. | General | $1 - 1/e$ | $1 - 1/e$ |
| Measured Greedy Frank-Wolfe | Monotone & non-neg. | Non-neg. | Down-closed | $1 - 1/e$ | $1/e$ |
| Measured Greedy Frank-Wolfe | Non-neg. | Monotone & non-neg. | Down-closed | $1/e$ | $1 - 1/e$ |
| Measured Greedy Frank-Wolfe | Monotone & non-neg. | Monotone & non-neg. | Down-closed | $1 - 1/e$ | $1 - 1/e$ |
| Measured Greedy Frank-Wolfe | Non-neg. | Non-neg. | Down-closed | $1/e$ | $1/e$ |
| Gradient Combining Frank-Wolfe | Monotone & non-neg. | General | General | $1/2 - \varepsilon$ | $1$ |
| Non-Oblivious Frank-Wolfe | Monotone & non-neg. | Non-neg. | General | $1 - 1/e - \varepsilon$ | $1 - \varepsilon$ |

Table 1: Summary of algorithms, settings, and guarantees ("NON-NEG." is a shorthand for "non-negative"). All of the conditions are in addition to the continuity and smoothness of $G$ and $C$, and the convexity of $P$. $\alpha$ and $\beta$ are the constants preceding $G(\mathbf{o})$ and $C(\mathbf{o})$ respectively in the lower bound on the output of the algorithm.

Our first basic assumption is that $G$ is non-negative. This assumption is necessary since we obtain multiplicative approximation guarantees with respect to $G$, and such guarantees do not make sense when $G$ is allowed to take negative values.[2] Our second basic assumption is that $P$ is solvable, i.e., that one can efficiently optimize linear functions subject to it. Intuitively, this assumption makes sense because one should not expect to be able to optimize a complex function such as $F$ subject to $P$ if one cannot optimize linear functions subject to it (nevertheless, it is possible to adapt our algorithms to obtain some guarantee even when linear functions can only be approximately optimized subject to $P$). Our final basic assumption is that both functions $G$ and $C$ are $L$-smooth, which means that they are differentiable, and moreover, their graidents are $L$-Lipschitz, i.e., $\|\nabla f(\mathbf{a}) - \nabla f(\mathbf{b})\|_2 \le L\|\mathbf{a} - \mathbf{b}\|_2 \quad \forall\, \mathbf{a}, \mathbf{b} \in [0,1]^n$ .

We conclude this section by introducing some additional notation that we need. We denote by $\mathbf{o}$ an arbitrary optimal solution for the problem described above, and define $D = \max_{\mathbf{x} \in P} \|\mathbf{x}\|_2$. Additionally, we denote by $\bar{0}$ and $\bar{1}$ the all-zeros and all-ones vector, respectively; and given two vectors $\mathbf{a}, \mathbf{b} \in \mathbb{R}^n$, we denote by $\mathbf{a} \odot \mathbf{b}$ their coordinate-wise multiplication, and by $\langle \mathbf{a}, \mathbf{b} \rangle$ their standard Euclidean inner product.

## 3 Main Algorithms and Results

In this section we present our (first-order) algorithms for solving the problem described in Section 2. In general, these algorithms are all Frank-Wolfe type algorithms, but they differ in the exact linear function which is maximized in each iteration (step 1 of the while/for loop), and in the formula used to update the solution (step 2 of the while/for loop). As mentioned previously, we assume everywhere that $G$ is a non-negative $L$-smooth DR-submodular function, $C$ is an $L$-smooth concave function, and $P$ is a solvable convex body. Some of our algorithms require additional non-negativity and/or monotonicity assumptions on the functions $G$ and $C$, and occasionally they also require a downward closed assumption on $P$. A summary of which settings each algorithm is applicable to can be found in Table 1. Each algorithm listed in the table outputs a point $x \in P$ which is guaranteed to obey $F(x) \ge \alpha \cdot G(\mathbf{o}) + \beta \cdot C(\mathbf{o}) - E$ for the constants $\alpha$ and $\beta$ given in Table 1 and some error term $E$.

**Remark:** An anonymous referee introduced us to a newly named class of functions called "up-concave" functions, which includes functions that are concave along non-negative directions (see [Wilder, 2018a]). It is known that both DR-submodular and concave functions are up-concave,

---

[2]We note that almost all the literature on submodular maximization of both discrete and continuous functions assumes non-negativity for the same reason.

| **Algorithm 1:** | **Algorithm 2:** |
| GREEDY FRANK-WOLFE $(\varepsilon)$ | MEASURED GREEDY FRANK-WOLFE $(\varepsilon)$ |
| --- | --- |
| Let $t \leftarrow 0$ and $\mathbf{y}^{(t)} \leftarrow \bar{0}$ | Let $t \leftarrow 0$ and $\mathbf{y}^{(t)} \leftarrow \bar{0}$ |
| **while** $t < 1$ **do** | **while** $t < 1$ **do** |
| $\quad \mathbf{s}^{(t)} \leftarrow \arg\max_{\mathbf{x} \in P} \langle \nabla F(\mathbf{y}^{(t)}), \mathbf{x} \rangle$ | $\quad \mathbf{s}^{(t)} \leftarrow \arg\max_{\mathbf{x} \in P} \langle (\bar{1} - \mathbf{y}^{(t)}) \odot \nabla F(\mathbf{y}^{(t)}), \mathbf{x} \rangle$ |
| $\quad \mathbf{y}^{(t+\varepsilon)} \leftarrow \mathbf{y}^{(t)} + \varepsilon \cdot \mathbf{s}^{(t)}$ | $\quad \mathbf{y}^{(t+\varepsilon)} \leftarrow \mathbf{y}^{(t)} + \varepsilon \cdot (\bar{1} - \mathbf{y}^{(t)}) \odot \mathbf{s}^{(t)}$ |
| $\quad t \leftarrow t + \varepsilon$ | $\quad t \leftarrow t + \varepsilon$ |
| **end while** | **end while** |
| **return** $\mathbf{y}^{(1)}$ | **return** $\mathbf{y}^{(1)}$ |

and therefore, the function $F$ we optimize also belongs to this class. More interestingly, all our results can be generalized (without any modification) to the case in which the function $G$ is an arbitrary non-negative up-concave function (rather than a non-negative DR-submodular function).

## 3.1 Greedy Frank-Wolfe Algorithm

In this section we assume that both $G$ and $C$ are monotone and non-negative functions (in addition to their other properties). Given this assumption, we analyze the guarantee of the greedy Frank-Wolfe variant appearing as Algorithm 1. This algorithm is related to the Continuous Greedy algorithm for discrete objective functions due to Călinescu et al. [2011], and it gets a quality control parameter $\varepsilon \in (0, 1)$. We assume in the algorithm that $\varepsilon^{-1}$ is an integer. This assumption is without loss of generality because, if $\varepsilon$ violates the assumption, then it can be replaced with a value from the range $[\varepsilon/2, \varepsilon]$ that obeys it. Most of the proofs of this section are deferred to Appendix A.1.

One can observe that the output $\mathbf{y}^{(1)}$ of Algorithm 1 is within the convex body $P$ because it is a convex combination of the vectors $\mathbf{s}^{(0)}, \mathbf{s}^{(\varepsilon)}, \mathbf{s}^{(2\varepsilon)}, \ldots, \mathbf{s}^{1-\varepsilon}$, which are all vectors in $P$. Let us now analyze the value of the output of Algorithm 1. The next lemma is the first step towards this goal. It provides a lower bounds on the increase in the value of $\mathbf{y}^{(t)}$ as a function of $t$.

**Lemma 3.1.** *For every* $0 \leq i < \varepsilon^{-1}$, $F(\mathbf{y}^{(\varepsilon(i+1))}) - F(\mathbf{y}^{(\varepsilon i)}) \geq \varepsilon \cdot [F(\mathbf{o}) - F(\mathbf{y}^{(\varepsilon i)})] - \varepsilon^2 L D^2$.

The corollary below follows by showing (via induction) that the inequality $F(\mathbf{y}^{(\varepsilon i)}) \geq [1 - (1 - \varepsilon)^i] \cdot F(\mathbf{o}) - i \cdot \varepsilon^2 L D^2$ holds for every integer $0 \leq i \leq \varepsilon^{-1}$, and then plugging in $i = \varepsilon^{-1}$. The inductive step is proven using Lemma 3.1.

**Corollary 3.2.** $F(\mathbf{y}^{(1)}) \geq (1 - e^{-1}) \cdot F(\mathbf{o}) - O(\varepsilon L D^2)$.

We are now ready to summarize the properties of Algorithm 1 in a theorem.

**Theorem 3.3.** *Let $S$ be the time it takes to find a point in $P$ maximizing a given liner function, then Algorithm 1 runs in $O(\varepsilon^{-1}(n + S))$ time, makes $O(1/\varepsilon)$ calls to the gradient oracle, and outputs a vector $\mathbf{y}$ such that $F(\mathbf{y}) \geq (1 - 1/e) \cdot F(\mathbf{o}) - O(\varepsilon L D^2)$.*

*Proof.* The output guarantee of Theorem 3.3 follows directly from Corollary 3.2. The time and oracle complexity follows by observing that the algorithm's **while** loop makes $\varepsilon^{-1}$ iterations, and each iteration requires $O(n + S)$ time, in addition to making a single call to the gradient oracle. $\square$

## 3.2 Measured Greedy Frank-Wolfe Algorithm

In this section we assume that $P$ is a down-closed body (in addition to being convex) and that $G$ and $C$ are both non-negative. Given these assumptions, we analyze the guarantee of the variant of Frank-Wolfe appearing as Algorithm 2, which is motivated by the Measured Continuous Greedy algorithm for discrete objective functions due to Feldman et al. [2011]. We again have a quality control parameter $\varepsilon \in (0, 1)$, and assume (without loss of generality) that $\varepsilon^{-1}$ is an integer.

The properties of Algorithm 2 are summarized in Theorem 3.4. Since the proof of this theorem can be viewed as a generalization of the proof of Theorem 3.3, we defer it to Appendix A.2.

**Theorem 3.4.** *Let $S$ be the time it takes to find a point in $P$ maximizing a given liner function, then Algorithm 2 runs in $O(\varepsilon^{-1}(n + S))$ time, makes $O(1/\varepsilon)$ calls to the gradient oracle, and outputs a vector $\mathbf{y}$ such that*

$$F(\mathbf{y}) \geq \begin{cases} 1 - e^{-1} & \text{if } G \text{ is monotone} \\ e^{-1} & \text{otherwise} \end{cases} \cdot G(\mathbf{o}) + \begin{cases} 1 - e^{-1} & \text{if } C \text{ is monotone} \\ e^{-1} & \text{otherwise} \end{cases} \cdot C(\mathbf{o}) - O(\varepsilon L D^2) \ .$$

### 3.3 Gradient Combining Frank-Wolfe Algorithm

Up to this point, the guarantees of the algorithms that we have seen had both $\alpha$ and $\beta$ that are strictly smaller than 1. However, since concave functions can be exactly maximized, it is reasonable to expect also algorithms for which the coefficient $\beta$ associated with $C(\mathbf{o})$ is equal to 1. In Sections 3.3 and 3.4, we describe such algorithms.

---

**Algorithm 3:** GRADIENT COMBINING FRANK-WOLFE $(\varepsilon)$

Let $\mathbf{y}^{(0)}$ be a vector in $P$ maximizing $C$ up to an error of $\eta \geq 0$.
**for** $i = 1$ **to** $\varepsilon^{-3}$ **do**
    $\mathbf{s}^{(i)} \leftarrow \arg\max_{\mathbf{x} \in P} \langle \nabla G(\mathbf{y}^{(i-1)}) + 2\nabla C(\mathbf{y}^{(i-1)}), \mathbf{x} \rangle$
    $\mathbf{y}^{(i)} \leftarrow (1 - \varepsilon^2) \cdot \mathbf{y}^{(i-1)} + \varepsilon^2 \cdot \mathbf{s}^{(i)}$
**end for**
**return** the vector maximizing $F$ among $\{\mathbf{y}^{(0)}, \ldots, \mathbf{y}^{(\varepsilon^{-3})}\}$

---

In this section, we assume that $G$ is a monotone and non-negative function (in addition to its other properties). The algorithm we study in this section is Algorithm 3, and it again takes a quality control parameter $\varepsilon \in (0, 1)$ as input. This time, however, the algorithm assumes that $\varepsilon^{-3}$ is an integer. As usual, if that is not the case, then $\varepsilon$ can be replaced with a value from the range $[\varepsilon/2, \varepsilon]$ that has this property. Most of the proofs of this section are deferred to Appendix A.3.

Firstly, note that for every integer $0 \leq i \leq \varepsilon^{-3}$, $\mathbf{y}^{(i)} \in P$; and therefore, the output of Algorithm 3 also belongs to $P$. For $i = 0$ this holds by the initialization of $\mathbf{y}^{(0)}$. For larger values of $i$, this follows by induction because $\mathbf{y}^{(i)}$ is a convex combination of $\mathbf{y}^{(i-1)}$ and the point $\mathbf{s}^{(i)}$ ($\mathbf{y}^{(i-1)}$ belongs to $P$ by the induction hypothesis, and $\mathbf{s}^{(i)}$ belongs to $P$ by definition).

Our next objective is to lower bound the value of the output point of Algorithm 3. For that purpose, it will be useful to define $\bar{F}(\mathbf{x}) = G(\mathbf{x}) + 2C(\mathbf{x})$ and $H(i) = \bar{F}(\mathbf{o}) - \bar{F}(\mathbf{y}^{(i)})$. To get a bound on the value of the output of Algorithm 3, we first show that $H(i)$ is small for at least some $i$ value. We do that using the next lemma, which shows that $H(i)$ decreases as a function of $i$ as longs as it is not already small compared to $G(\mathbf{y}^i)$. Then, Corollary 3.6 guarantees the existence of a good iteration $i^*$.

**Lemma 3.5.** *For every integer $1 \leq i \leq \varepsilon^{-3}$, $H(i-1) - H(i) \geq \varepsilon^2 \cdot [G(\mathbf{o}) - 2G(\mathbf{y}^{(i-1)})] + 2\varepsilon^2 \cdot [C(\mathbf{o}) - C(\mathbf{y}^{(i-1)})] - 6\varepsilon^4 L D^2 = \varepsilon^2 \cdot [H(i-1) - G(\mathbf{y}^{(i-1)})] - 6\varepsilon^4 L D^2.$*

**Corollary 3.6.** *There is an integer $0 \leq i^* \leq \varepsilon^{-3}$ obeying $H(i^*) \leq G(\mathbf{y}^{(i^*)}) + \varepsilon \cdot [G(\mathbf{o}) + 2\eta + 6L D^2].$*

We are now ready to summarize the properties of Algorithm 3 in a theorem.

**Theorem 3.7.** *Let $S_1$ be the time it takes to find a point in $P$ maximizing a given liner function and $S_2$ be the time it takes to find a point in $P$ maximizing $C(\cdot)$ up to an error of $\eta$, then Algorithm 2 runs in $O(\varepsilon^{-3} \cdot (n + S_1) + S_2)$ time, makes $O(1/\varepsilon^3)$ gradient oracle calls, and outputs a vector $\mathbf{y}$ such that*

$$F(\mathbf{y}) \geq \tfrac{1}{2}(1 - \varepsilon) \cdot G(\mathbf{o}) + C(\mathbf{o}) - \varepsilon \cdot O(\eta + L D^2) \ .$$

*Proof.* We begin the proof by analyzing the time and oracle complexities of Algorithm 3. Every iteration of the loop of Algorithm 3 takes $O(n + S_1)$ time. As there are $\varepsilon^{-3}$ such iterations, the entire algorithm runs in $O(\varepsilon^{-3}(n + S_1) + S_2)$ time. Also note that each iteration of the loop requires 2 calls to the gradient oracles (a single call to the oracle corresponding to $G$, and a single call to the oracle corresponding to $C$), so the overall oracle complexity of the algorithm is $O(1/\varepsilon^3)$.

Consider now iteration $i^*$, whose existence is guaranteed by Corollary 3.6. Then,

$$H(i^*) \leq G(\mathbf{y}^{(i^*)}) + \varepsilon \cdot [G(\mathbf{o}) + 2\eta + 6LD^2]$$
$$\implies [G(\mathbf{o}) + 2C(\mathbf{o})] - [G(\mathbf{y}^{(i^*)}) + 2C(\mathbf{y}^{(i^*)})] \leq G(\mathbf{y}^{(i^*)}) + \varepsilon \cdot [G(\mathbf{o}) + 2\eta + 6LD^2]$$
$$\implies F(\mathbf{y}^{(i^*)}) \geq \tfrac{1}{2}(1 - \varepsilon) \cdot G(\mathbf{o}) + C(\mathbf{o}) - \varepsilon \cdot [\eta + 3LD^2] \ .$$

The theorem now follows since the output of Algorithm 2 is at least as good as $\mathbf{y}^{(i^*)}$. $\qquad\square$

### 3.4 Non-oblivious Frank-Wolfe Algorithm

As mentioned in the beginning of Section 3.3, our objective in this section is to present another algorithm that has $\beta = 1$ (*i.e.*, it maximizes $C$ "exactly" in some sense). In Section 3.3, we presented Algorithm 3, which achieves this goal with $\alpha = 1/2$. The algorithm we present in the current section achieves the same goal

---
**Algorithm 4:** NON-OBLIVIOUS FRANK-WOLFE $(\varepsilon)$

Let $\mathbf{y}^{(0)}$ be an arbitrary vector in $P$, and let $\beta(\varepsilon) \leftarrow e(1 - \ln \varepsilon)$.
**for** $i = 0$ **to** $\lceil e^{-1} \cdot \beta(\varepsilon)/\varepsilon^2 \rceil$ **do**
  $\mathbf{s}^{(i)} \leftarrow \arg\max_{\mathbf{x} \in P} \langle e^{-1} \cdot \nabla \bar{G}(\mathbf{y}^{(i)}) + \nabla C(\mathbf{y}^{(i)}), \mathbf{x} \rangle$
  $\mathbf{y}^{(i+1)} \leftarrow (1 - \varepsilon) \cdot \mathbf{y}^{(i)} + \varepsilon \cdot \mathbf{s}^{(i)}$
**end for**
**return** the vector maximizing $F$ among $\{\mathbf{y}^{(0)}, ..., \mathbf{y}^{(\lceil e^{-1} \cdot \frac{\beta(\varepsilon)}{\varepsilon^2} \rceil)}\}$

---

with an improved value of $1 - 1/e$ for $\alpha$. However, the improvement is obtained at the cost of requiring the function $C$ to be non-negative (which was not required in Section 3.3). Additionally, like in the previous section, we assume here that $G$ is a monotone and non-negative function (in addition to its other properties).

The algorithm we study in this section is a non-oblivious variant of the Frank-Wolfe algorithm, appearing as Algorithm 4, which takes a quality control parameter $\varepsilon \in (0, 1/4)$ as input. As usual, we assume without loss of generality that $\varepsilon^{-1}$ is an integer. Algorithm 4 also employs the non-negative auxiliary function: $\bar{G}(\mathbf{x}) = \varepsilon \cdot \sum_{j=1}^{\varepsilon^{-1}} \frac{e^{\varepsilon j} \cdot G(\varepsilon j \cdot \mathbf{x})}{\varepsilon j}$. This function is inspired by the non-oblivious objective function used by Filmus and Ward [2012].

Note that any call to the gradient oracle of $\bar{G}$ can be simulated using $\varepsilon^{-1}$ calls to the gradient oracle of $G$. The properties of Algorithm 4 are stated in Theorem 3.8. Since the proof of this theorem is quite technical and reuses some of ideas from the proof of Theorem 3.7 (as well as new ideas), we defer it to Appendix A.4.

**Theorem 3.8.** *Let $S$ be the time it takes to find a point in $P$ maximizing a given linear function, then Algorithm 4 runs in $O(\varepsilon^{-2}(n/\varepsilon + S) \ln \varepsilon^{-1})$ time, makes $O(\varepsilon^{-3} \ln \varepsilon^{-1})$ gradient oracle calls, and outputs a vector $\mathbf{y}$ such that:*

$$F(\mathbf{y}) \geq (1 - 1/e - 4\varepsilon \ln \varepsilon^{-1}) \cdot G(\mathbf{o}) + (1 - 4\varepsilon \ln \varepsilon^{-1}) \cdot C(\mathbf{o}) - 4\varepsilon LD^2 \ .$$

## 4 Experiments

In this section we describe some experiments pertaining to our algorithms for maximizing DR-submodular + concave functions. All experiments are done on a 2015 Apple MacBook Pro with a quad-core 2.2 GHz i7 processor and 16 GB of RAM.

### 4.1 Interpolating Between Constrasting Objectives

We use our algorithms for maximizing the sum of a DR-submodular function and a concave function to provide a way to achieve a trade-off between different objectives. For example, given a ground set $X$ and a DPP supported on the power set $2^X$, the maximum a posteriori (MAP) of the DPP corresponds to picking the most likely (diverse) set of elements according to the DPP. On the other hand, concave functions can be used to encourage points being closer together and clustered.

Finding the MAP of a DPP is an NP-hard problem. However, continuous approaches employing the multilinear extension [Călinescu et al., 2011] or the softmax extension [Bian et al., 2017a, Gillenwater et al., 2012] provide strong approximation guarantees for it. The softmax approach is usually preferred as it has a closed form solution which is easier to work with. Now, suppose

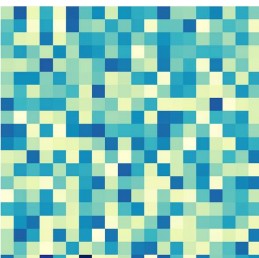 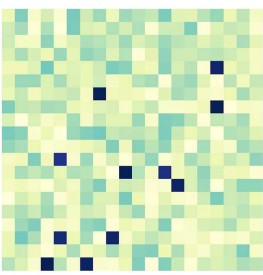 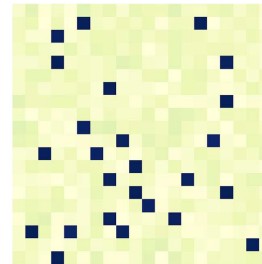

Figure 2: Outputs of the experiment described in Section 4.1. LEFT PLOT: $\lambda = 1$ (just the submodular objective). RIGHT PLOT: $\lambda = 0$ (just the concave objective). MIDDLE PLOT: $0 < \lambda < 1$ (combination of both objectives). A darker square corresponds to a larger entry in the final output of Algorithm 3 for the corresponding element.

that $|X| = n$, and let $\mathbf{L}$ be the $n \times n$ kernel of the DPP and $\mathbf{I}$ be the $n \times n$ identity matrix, then $G(\mathbf{x}) = \log \det[\operatorname{diag}(\mathbf{x})(\mathbf{L} - \mathbf{I}) + \mathbf{I}]$ is the softmax extension for $\mathbf{x} \in [0, 1]^n$. Here, $\operatorname{diag}(\mathbf{x})$ corresponds to a diagonal matrix with the entries of $\mathbf{x}$ along its diagonal.

Observe now that, given a vector $\mathbf{x} \in [0, 1]^n$, $x_i$ can be thought of as the likelihood of picking element $i$. Moreover, $\mathbf{L}_{ij}$ captures the similarity between elements $i$ and $j$. Hence, our choice for a concave function which promotes similarity among elements is $C(\mathbf{x}) = \sum_{i,j} \mathbf{L}_{ij}(1 - (x_i - x_j)^2)$. The rationale behind this is as follows. For a particular pair of elements $i$ and $j$, if $\mathbf{L}_{ij}$ is large, that means that $i$ and $j$ are similar, so we would want $C$ to be larger when $\mathbf{L}_{ij}$ is high, provided that we are indeed *picking both* $i$ and $j$ (i.e., provided that $(x_i - x_j)^2$ is small). One can verify that the function $C(\mathbf{x})$ is indeed concave as its Hessian is negative semidefinite.

In our first experiment we fix the ground set to be the set of $20 \times 20 = 400$ points evenly spaced in $[0, 1] \times [0, 1] \subset \mathbb{R}^2$. We also choose $\mathbf{L}$ to be the Gaussian kernel $\mathbf{L}_{ij} = \exp(-d(i, j)^2 / 2\sigma^2)$, where $d(i, j)$ is the Euclidean distance between points $i$ and $j$, and $\sigma = 0.04$. Given the functions $G$ and $C$ defined above, we optimize in this experiment a combined objective formally specified by $F = \lambda G + (1 - \lambda)C$, where $\lambda \in [0, 1]$ is a control parameter that can be used to balance the contrasting objectives represented by $G$ and $C$. For example, setting $\lambda = 1$ produces the (spread out) pure DPP MAP solution, setting $\lambda = 0$ produces the (clustered) pure concave solution and $\lambda = 0.5$ produces a solution that takes both constraints into consideration to some extent. It is important to note, however, that the effect of changing $\lambda$ on the importance that each type of constraint gets is not necessarily linear—although it becomes linear when the ranges of $G$ and $C$ happen to be similar.

In Figure 2, we can see how changing $\lambda$ changes the solution. The plots in the figure show the final output of Algorithm 3 when run on just the submodular objective $G$ (left plot), just the concave objective $C$ (right plot), and a combination of both (middle plot). The algorithm is run with the same cardinality constraint of 25 in all plots, which corresponds to imposing that the $\ell_1$ norm of each iteration must be at most 25. It is important to note that we represent the exact (continuous) output of the algorithm here. To get a discrete solution, a rounding method should be applied. Also, all of the runs of the algorithm begin from the same fixed starting point inside the cardinality constrained polytope. The step sizes used by the different runs are all constant, and were chosen empirically.

**On Jeopardy Dataset.** As we can see, the above setup lends itself nicely to interpolating between contrasting objectives, and this can be used in real world applications. For example, consider a set of news articles or sentences. The DPP MAP solution would correspond to picking a very diverse set of articles, whereas using just the concave objective would pick a set of similar articles or sentences. The important step is to appropriately codify the similarity between elements of the ground set (i.e. the news articles or the sentences) in the kernel $\mathbf{L}$. To illustrate this, we consider a set of 324 questions from the game show Jeopardy (taken from https://www.j-archive.com/), use a pretrained BERT model to map these questions to (BERT) embeddings, and then use cosine-similarity to construct the matrix $\mathbf{L}$. This matrix can be seen to be positive semidefinite. Though we do not have a (small) set of topics or meta-features for these questions, when this experiment is run, we get that the concave solution picks questions focused on a few topics such as history and fantasy (like *"Star designer John Galliano was born Juan Carlos Galliano in this British possession at the tip of Spain"*, *"In 1801 this onetime VP compiled 'A Manual of Parliamentary Practice' still used in the U.S. Senate"*,

and *"In Euripides' play about this famed beauty, it's her double who goes to Troy"*), whereas the DPP solution is much more spread out (including questions such as *"These parts of a peach tree are glossy green, pointed and lance shaped"* and *"5 x 10 x 15"*).

## 4.2   Other Submodular + Concave Objectives

In this section we compare our algorithms with two baselines: the Frank-Wolfe algorithm [Frank and Wolfe, 1956] and the projected gradient ascent algorithm. In all the experiments done in this section, the objective function is again $F = \lambda G + (1 - \lambda)C$, where $G$ and $C$ are a DR-submodular function and a concave function, respectively. For simplicity, we set $\lambda = 1/2$ throughout the section.

### 4.2.1   Quadratic Programming

The non-convex quadratic programming (NQP) problem is a fundamental NP-hard problem studied within the realm of DR-submodular quadratic programming in works such as Bian et al. [2017a] and on its own [Xia et al., 2018, Burer and Letchford, 2009, Sherali and Tuncbilek, 1995]. Here, we use our submodular + concave framework to study the NQP problem for a class of Hessian matrices which can be decomposed as the sum of a Hessian corresponding to a submodular objective and a Hessian corresponding to a concave objective. We define $G(\mathbf{x}) = \frac{1}{2} \mathbf{x}^\top \mathbf{H} \mathbf{x} + \mathbf{h}^\top \mathbf{x} + c$. By choosing the matrix $\mathbf{H}$ and vector $\mathbf{h}$ appropriately, this objective can be made to be monotone or non-monotone DR-submodular. We also define the down-closed constraint set to be $P = \{\mathbf{x} \in \mathbb{R}_+^n \mid \mathbf{A}\mathbf{x} \leq \mathbf{b}, \mathbf{x} \leq \mathbf{u}, \mathbf{A} \in \mathbb{R}_+^{m \times n}, \mathbf{b} \in \mathbb{R}_+^m\}$. Following Bian et al. [2017a], we choose the matrix $\mathbf{H} \in \mathbb{R}^{n \times n}$ to be a randomly generated symmetric matrix with entries uniformly distributed in $[-1, 0]$, and the matrix $\mathbf{A}$ to be a random matrix with entries uniformly distributed in $[0.01, 1.01]$ (the $0.01$ addition here is used to ensure that the entries are all positive). The vector $\mathbf{b}$ is chosen as the all ones vector, and the vector $\mathbf{u}$ is a tight upper bound on $P$ whose $i^{\text{th}}$ coordinate is defined as $u_i = \min_{j \in [m]} b_j / \mathbf{A}_{ji}$. We let $\mathbf{h} = -0.2 \cdot \mathbf{H}^\top \mathbf{u}$ which makes $G$ non-monotone. Finally, although this does not affect the results of our experiments, we take $c$ to be a large enough additive constant (in this case 10) to make $G$ non-negative.

It is known that when the Hessian of a quadratic program is negative semidefinite, the resulting objective is concave. Accordingly, we let $C(\mathbf{x}) = \frac{1}{20} \mathbf{x}^\top \mathbf{D} \mathbf{x}$, where $\mathbf{D}$ is a negative semidefinite matrix defined by the negation of the product of a random matrix with entries in $[0, 1]$ with its transpose. As one can observe, the generality of DR-submodular + concave objectives allows us to consider quadratic programming with very different Hessians. We hope that our ability to do this will motivate future work about quadratic programming for a broader class of Hessian matrices.

In the current experiment, we let $n \in \{8, 12, 16\}$ and $m \in \{0.5n, n, 1.5n\}$, and run each algorithm for 50 iterations. Note that having fixed the number of iterations, the maximum step size for Algorithms 1 and 2 is upper bounded by (number of iterations)$^{-1}$ = $1/50$ to guarantee that these algorithms remain within the polytope. To ensure consistency, we set the step sizes for the other algorithms to be $1/50$ as well, except for Algorithm 4 for which we set to the value of $\varepsilon$ given by solving $e^{-1} \cdot \beta(\varepsilon)/\varepsilon^2 = 50$. This ensures that the gradient computation in Algorithm 4 is not too time consuming. We start Algorithms 1 and 2 from the starting point their pseudocodes specify, and the other algorithms from the same arbitrary point. We show the results for $n = 8$ and $m = 4, 8, 12$ in Figures 3a, 3b, and 3c, respectively (each plot shows the average of 50 runs of the experiment). Due to space constraints, the results for $n = 12$ and 16 are postponed to Appendix B. We also note that since Algorithms 3 and 4 output the best among the results of all their iterations, we just plot the final output of these algorithms instead of the entire trajectory.

### 4.2.2   D-optimal Experimental Design

Following Chen et al. [2018], the DR-submodular objective function for the D-optimal experimental design problem is $G(\mathbf{x}) = \log \det\left(\sum_{i=1}^n x_i \mathbf{Y_i}^\top \mathbf{Y_i}\right)$. Here, $\mathbf{Y_i}$ is an $n$ dimensional row-vector in which each entry is drawn independently from the standard Gaussian distribution. The choice of concave function is $C(\mathbf{x}) = \frac{1}{10} \sum_{i=1}^n \log(x_i)$. In this experiment there is no combinatorial constraint. Instead, we are interested in maximization over a box constraint, i.e., over $[1, 2]^n$ (note that the box is shifted compared to the standard $[0, 1]^n$ to ensure that $G$ is defined everywhere as it is undefined at $\mathbf{x} = \mathbf{0}$). The final outputs of all the algorithms for $n = 8, 12, 16$ appear in Figure 3d. Like in

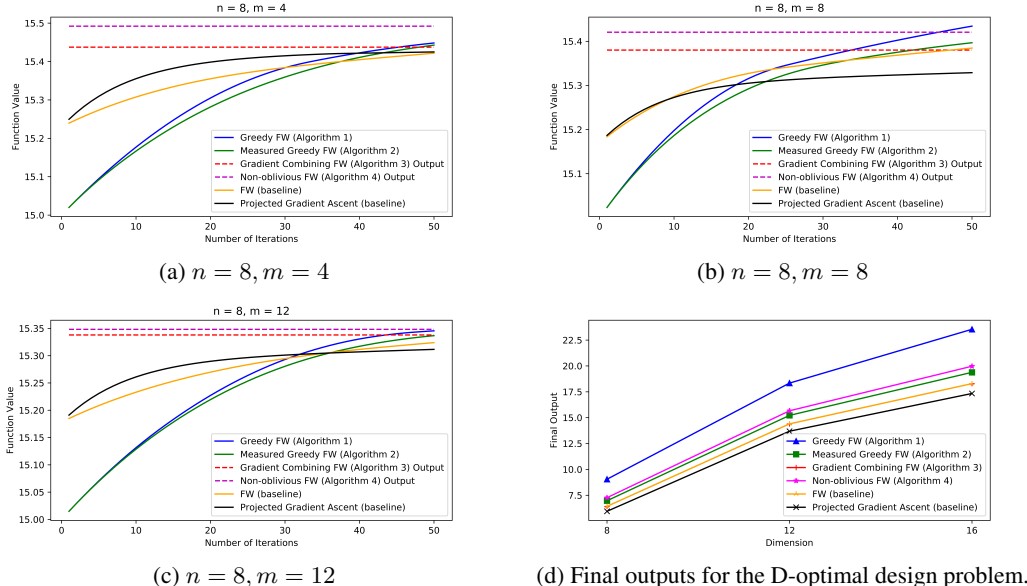

(a) $n = 8, m = 4$

(b) $n = 8, m = 8$

(c) $n = 8, m = 12$

(d) Final outputs for the D-optimal design problem.

Figure 3: Plots 3a, 3b, and 3c correspond to the quadratic programming experiment from Section 4.2.1. Plot 3d pertains to the D-optimal experimental design problem from Section 4.2.2.

Section 4.2.1, each algorithm was run for $50$ iterations, and each plot is the average of $50$ runs. The step sizes and starting points used by the algorithms are set exactly like in Section 4.2.1.

**Takeaways.** Based on our experiments, we can observe that Algorithms 1 and 4 consistently outperform the other algorithms. We can also see (especially in the D-optimal experimental design problem where they almost superimpose) that the difference between Algorithm 3 and the standard Frank-Wolfe algorithm are minimal, but we believe that the difference between the two algorithms can be made more pronounced by considering settings in which the gradient of $C$ dominates the gradient of $G$. Finally, one can note that the output value in plots 3a, 3b, and 3c tends to decrease when the number of constraints increases, which matches our intuitive expectation.

## 5 Conclusion

In this paper, we have considered the maximization of a class of objective functions that is strictly larger than both DR-submodular functions and concave functions. The ability to optimize this class of functions using first-order information is interesting from both theoretical and practical points of view. Our results provide a step towards the goal of efficiently analyzing structured non-convex functions—a goal that is becoming increasingly relevant.

Our various results enable us to obtain approximation guarantees for many use cases. One use case which is somewhat less well-covered by our results is the case of maximizing the difference between a non-negative monotone DR-submodular function $G$ and a non-negative monotone convex function $C$. Currently, only Algorithm 3 applies to this case, yielding an approximation guarantee of $\alpha = 1/2$ and $\beta = 1$ (in the notation of Table 1). However, as the functions in this case have so many properties on top of what is required by Algorithm 3, it is reasonable to believe that one can improve over this approximation guarantee.

## Funding Transparency Statement

**Funding in direct support of this work:** The work of Moran Feldman was supported in part by Israel Science Foundation (ISF) grant no. 459/20. Amin Karbasi acknowledges funding in direct support of this work from NSF (IIS-1845032) and ONR (N00014-19-1-2406).

**Additional revenues related to this work:** None

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
