# OpenReview forum: "Submodular + Concave"
_NeurIPS.cc/2021/Conference — NeurIPS 2021 Poster_

### Official Review · Reviewer_rEMg · 2021-06-25

**Rating:** 6
**Confidence:** 3

**Summary:**

This paper studies maximization of submodular + concave functions over solvable polytopes. This problem forms a new class of non-convex optimization problems, for which no theoretical results have been obtained. Depending on the conditions of the problem, the authors develop several algorithms and prove that they return approximately optimal solutions. Experiments demonstrate the practical utility of the proposed algorithms.

--- After rebuttal ---

I appreciate the authors' kind response. My technical concerns have been resolved. However, I still think the experimental comparison in running time is important to clearly demonstrate the effectiveness of Algorithm 4 since $\nabla \bar{G}$ usually requires more computation costs than $\nabla \bar{G}$, as mentioned in the response. Despite the lack of experiments, the theoretical contributions are great and I think the paper is above the acceptance threshold. Therefore, I keep my score of 6.

**Limitations And Societal Impact:**

The limitations are described. The paper seems to have no negative societal impacts.

**Main Review:**

The paper addresses a new class of optimization problems, which are motivated by some machine learning applications (e.g., DPP + clustered). The theoretical guarantees are reasonable and strong. Overall, I think this is a good paper. There are, however, some concerns and comments as listed below:

1. Since each algorithm and its analysis are explained only briefly, it is hard to see how novel and significant each of them is. Algorithms 1 and 2 seem to be similar to continuous greedy and measured continuous greedy. Regarding the analyses of them, are there some special techniques required to deal with the additional concave function? Regarding Algorithms 3 and 4, are there existing studies that provide their bases, or are they newly developed for submodular + concave maximization?

2. I would like the authors to show how to guarantee outputs of Algorithm 2 are included in $P$.

3. I think the class of quadratic functions considered in Section 4.2.1 is interesting. I would appreciate it if the authors could present relevant previous studies on non-convex quadratic programming, if any, and explain advantages of the submodular + concave framework in the quadratic-programming context.

4. In Figure 3, algorithms are compared based on the number of iterations. What if those are compared based on running times?

Minor comments:

- It is better to define $\bar 0$ and $\bar 1$ in Section 2.

**Time Spent Reviewing:**

3

---

> ### Author Response · Authors · 2021-08-06
> **Response to Review**
>
> We thank the reviewer for their response. Algorithm 4 is related to the work of Filmus and Ward (2012) (this is mentioned on Page 7), while Algorithm 3 can be viewed as a counterpart of the standard discrete local search algorithm. As mentioned in the last paragraph in response to the comments of Reviewer 3, the crux of our contribution is the ability to apply algorithms from submodular maximization in parallel to optimizing a concave function. In Algorithms 1 and 2 this is relatively easy, but is much more involved in Algorithms 3 and 4.
>
> The fact that the output of Algorithm 2 is contained in P is mentioned in the submconcave_full.pdf file in the Supplementary Material and can be seen in line 584 as a consequence of Lemma A.1. Specifically, the output of Algorithm 2 is (coordinate wise) upper bounded by a convex combination of points in $P$ (which is a point in $P$), and then the down-closed nature of $P$ is used to ensure that the output is also in $P$.
>
> Non-convex quadratic programming (NQP) has been discussed in several works such as “Guaranteed Non-convex Optimization:Submodular Maximization over Continuous Domains” by Bian, Mirzasoleiman, Buhmann, and Krause. They consider randomly generated DR-submodular objectives within the class of NQP problems. Our submodular + concave framework allows us to extend this and optimize a larger class of NQP instances.
>
> Though we have not compared the algorithms based on runtimes, we believe they should all be qualitatively similar. Algorithms 1, 2, and 3 and the baseline algorithms follow a similar template and their runtimes should be very similar. Algorithm 4 introduces an auxiliary function ($\bar{G}$). If we are provided with a gradient oracle directly for $\bar{G}$, then the runtime for Algorithm 4 should also be similar to that of the others. However, if we use an oracle for $G$ to implement an oracle for $\bar{G}$, then Algorithm 4 would be slower. We avoid comparing the algorithms in the paper based on their runtimes because such a comparison is prone to technical factors such as the exact coding of the algorithms and the details of the system’s architecture, which we want to avoid.

---

### Official Review · Reviewer_mN3h · 2021-07-08

**Rating:** 6
**Confidence:** 5

**Summary:**

This paper studies a class of optimization problems where the goal is to maximize the sum of a DR-submodular function and a concave function. Depending on whether the DR-submodular function is monotone, the concave function is monotone or non-negative, and the constraint set is down-closed, the authors have proposed 4 algorithms with various approximation ratios for the DR-submodular and concave parts of the objective function. Finally, this work provides a set of experiments to compare the performance of the proposed algorithms with baseline algorithms for DR-submodular maximization and concave maximization.

**Limitations And Societal Impact:**

No, however, I don't think neither was really needed to be addressed. In terms of limitations, Table 1 clearly highlights the settings that each of the proposed algorithms could be applied to, and therefore, it is easy to infer the problems that are not addressed by this work. Also, this paper is mainly a theoretical work and does not raise any potential ethical concerns.

**Main Review:**

The paper has many strengths that are listed below:
- Theoretical analysis of the previously unstudied framework of DR-submodular + concave maximization and providing new algorithms (inspired by the algorithms for DR-submodular maximization) that obtain separate approximation ratios for the DR-submodular and the concave part.
- Comprehensive set of numerical experiments to highlight the performance of the proposed algorithms and compare/contrast them with the previously known algorithms for either DR-submodular maximization or concave maximization.
- Great review of all the previous works that are either studying a similar framework or using similar techniques and ideas.
- Easy to follow proofs (in the appendix) for all the claims and results of the paper.

On the other hand, this work has some limitations as follows:
- In my view, the framework of DR-submodular + concave maximization is not well motivated in the paper. While the DR-submodular + linear setting is well-motivated in prior works, I'm not sure how important this extension to concave functions is (aside from the additional theoretical challenges to obtain results). It seems like the chosen concave functions in all the provided experiments are also arbitrarily chosen and not inspired by a practical application.
- One of the most important subclasses of this framework is "monotone non-negative DR-submodular - monotone non-negative convex" maximization which is a penalized formulation of constrained DR-submodular maximization with convex constraints. Out of the 4 proposed algorithms, the Gradient Combining Frank-Wolfe algorithm (with approximation ratios 1/2 and 1 for the DR-submodular and concave parts respectively) is the only one that could be used for this problem. Given the specific structure of the concave function in this example (negative and monotone decreasing), I think it is possible to obtain a better approximation ratio for the DR-submodular part.
- Most of the ideas for designing the proposed algorithms are heavily inspired by prior works on submodular maximization and although the theoretical analysis for the DR-submodular + concave is not exactly similar to prior results, it could be considered a straightforward extension of previous proofs (particularly for the first two proposed algorithms).
**********************************
Post-Rebuttal Update: Thanks to the authors for their careful responses to the raised questions. I totally agree that "the addition of a concave function can be seen as the usage of a concave regularizer for the maximization of DR-submodular functions" and this is basically the "monotone non-negative DR-submodular - monotone non-negative convex" case that I mentioned earlier. For instance, a knapsack or budget constraint $c^Tx\leq 1$ could be enforced in the penalized form as $C(x)=-P(c^Tx)$ where $P$ is a monotone and convex function (e.g., $P(z)=exp(z)$). I think a separate discussion of this framework and obtaining an improved approximation ratio $\alpha>\frac{1}{2}$ for this special case would be interesting.

**Time Spent Reviewing:**

4-5 hours to read the paper and also almost all of the proofs provided in the appendix.

---

> ### Author Response · Authors · 2021-08-06
> **Response to Review**
>
> We thank the reviewer for their feedback. Regarding the motivations for studying this framework, we have mentioned a few examples in the paper. This framework allows us to study non-unimodal distributions which are becoming increasingly important given that many distributions we come across in ML tasks are multimodal. Secondly, the addition of a concave function can be seen as the usage of a concave regularizer for the maximization of DR-submodular functions, which has traditionally been studied with just linear regularization. Lastly, the experiment in Section 4.1 allows us to contrast between competing objectives (spread out vs. clustered), and striking a balance between these opposing objectives may be of interest in certain applications (for example, say you want to select some news articles out of a corpus of articles, but want them to span 3 or 4 different topics as opposed to all the articles belonging to the same topic, or all belonging to different topics.)  As far as the specific choice of concave functions are concerned, we motivate the chosen function for the experiment in Section 4.1 in the paragraph beginning on line 274; and in Section 4.2.1, we take the concave function to be a quadratic program with a (randomly generated) negative semidefinite hessian as this seems quite natural for the quadratic programming experiment.
>
> Regarding the monotone non-negative DR-submodular - monotone non-negative concave maximization problem. In the special case where $C$ is linear and non-positive, standard techniques can be used to obtain $\alpha = 1-1/e$ and $\beta = 1$, and we will add a short remark discussing this. However, we currently do not know how to get similar factors for a general $C$ functions of the kind mentioned by the reviewer.
>
> Regarding the reliance on prior proof techniques, our work does use many existing ideas from the literature on submodular functions. However, the challenge was to make these ideas apply to $G$ while simultaneously being able to analyze the guarantee for $C$. In some cases (like the first two algorithms) this was relatively straightforward, but in other cases this was much more difficult. For example, our last two algorithms have to return the best of all the intermediate solutions they encounter because the interplay between $G$ and $C$ makes it impossible to guarantee convergence to a good solution. Instead, we have to argue that somewhere along the way there is a solution that behaves similarly to a local optimum with respect to $G$.

---

### Official Review · Reviewer_8aFc · 2021-07-15

**Rating:** 7
**Confidence:** 4

**Summary:**

This paper studies the maximization of a nonconvex function in the form of $F = G + C$ on a convex set $P \\subseteq [0,1]^n$, where $G$ is DR-submodular and $C$ is concave. For various settings (monotonicity/nonnegativity of $G$ or $C$, $P$ is a down-closed/general convex set, the gradient oracle is for only $F$/both $G$ and $C$, etc.), the authors developed the first approximation algorithms.

The algorithms exploit some ideas from submodular function maximization such as Frank-Wolfe, measured continuous greedy, etc. The algorithms have provable fine approximation guarantees, i.e., they achieve $\\alpha$- and $\\beta$-approximation for $G$ and $C$, respectively. These are stronger guarantees than the standard approximation results in submodular function maximization. Furthermore, the authors proved even tighter approximation ratios (i.e., $\\beta \\approx 1$) if the individual gradients of $G$ and $C$ are available.

The authors conducted the numerical experiments in (nonconvex) quadratic programming and D-optimal design. The proposed algorithms achieved better objective values compared to the baseline methods.

**Ethics Review Area:**

["I don’t know"]

**Main Review:**

Overall, I enjoyed reading this paper. The class of nonconvex functions is reasonable and interesting because we can obtain constant-factor approximation guarantees. The paper is well-structured and easy to follow (if one is familiar with submodular/convex analysis). The authors provided effective algorithms for various settings. Each proposed algorithm nicely extends the known algorithm for submodular or concave maximization. The experiment with real-world data shows the efficacy of the proposed algorithms. In summary, the paper made significant theoretical contributions to nonconvex optimization using submodular maximization techniques, and the results are supported by real-world experiments. I would recommend accepting it.

## Minor comment
The class of submodular+concave is contained in the class of up-concave functions (i.e., concave along nonnegative directions). Can we understand the results from the up-concavity of objective functions? I guess Greedy Frank-Wolfe also works for up-concave functions with the same approximation ($\\alpha=\\beta=1-1/e$), but not sure for the rest. It would be helpful if the authors have comments on the relation to up-concavity.

**Time Spent Reviewing:**

5

---

> ### Author Response · Authors · 2021-08-06
> **Response to Review**
>
> We thank the reviewer for their comments. We are not familiar with the class of up-concave functions and would appreciate it if the reviewer could point us to some references so that we can refer to this subject in the revised version of the paper. It is true that the entire objective function in our setting (i.e., $G + C$) is an up-concave function, but it might be more useful to ask whether our results hold when the requirement that $G$ is DR-submodular is replaced with the weaker requirement that $G$ is an up-concave function (note that this setting is strictly more powerful than simply assuming that $G + C$ is up-concave since $C$ can always be chosen as the zero function). Interestingly, it turns out that all our results indeed apply also when $G$ is only guaranteed to be up-concave instead of DR-submodular, and moreover, there is no need to modify our proofs at all to adapt them to this more general setting.

---

> > ### Comment · Reviewer_8aFc · 2021-08-23
> > **Reference on up-concavity**
> >
> > Here is some reference which explicitly used up-concavity beyond DR-submodularity. The author applied Frank-Wolfe to some up-concave function appearing in CVaR optimization to get $(1-1/e)$-approximation. I hope this helps.
> >
> > Wilder, B. (2018). Risk-Sensitive Submodular Optimization. Proceedings of the AAAI Conference on Artificial Intelligence, 32(1).
> > https://ojs.aaai.org/index.php/AAAI/article/view/12121

---

> > > ### Author Response · Authors · 2021-08-25
> > > **Response to Comment**
> > >
> > > Thank you for the pointer. We will take a look.

---

### Official Review · Reviewer_aSDv · 2021-07-17

**Rating:** 7
**Confidence:** 4

**Summary:**

Authors consider the problem maximizing a function that can be written as the addition of submodular and concave functions. They provide multiple Frank-Wolfe-based algorithms for this problem with different approximation factors.

**Limitations And Societal Impact:**

The authors discussed the limitations and potential negative social impact of their work.



**Main Review:**

1)Author motivates the subject from a theoretical and practical point of view. But all the experiments are on synthetic data. It would be great if they can add simulation for an application on a real dataset.

2)In the introduction, the author reviewed the previous works carefully.

3)the author successfully addressed the issue and recommended the algorithms. They proposed this new problem. Their idea to change the frank Wolfe algorithm to come up with a better approximation factor is interesting. Although, if they can characterize what type of functions can be written as sub+concave or, for example, come up with the condition on second-order derivative, which is equivalent to writing a function as sub+concave that would be very interesting.

4)The authors successfully provide simulation results for their algorithm.

5)The paper is well-organized and well-written.

6)The proof is mathematically correct.

7)the code is included.

Overall, the idea is interesting, and the author's way of solving the problem is interesting too.


----------- Post Rebuttal-------------

I have read all the comments, and I don't change my score.

**Time Spent Reviewing:**

6

---

> ### Author Response · Authors · 2021-08-06
> **Response to Questions**
>
> We thank the reviewer for carefully reading the paper. We will add an experiment on real world data in the final version.
>
> Since a twice differentiable function is DR-submodular iff its hessian is non-positive at every point in the domain and since a twice differentiable function is concave iff its hessian is negative semidefinite, we can say that a function belongs to the submodular+concave class iff its hessian can be decomposed as the sum of a matrix which is non-positive everywhere and a matrix which is negative semidefinite.

---

### Decision · Program_Chairs · 2021-09-27

**Decision:**

Accept (Poster)

**Comment:**

The reviewers agree that this generally a good paper although not entirely without (minor) flaws. Please take the reviewers comments in consideration when preparing a revision. The answers provided by the authors were given due consideration.